# Environmental Noise Exposure and Sleep Habits among Children in a Cohort from Northern Spain

**DOI:** 10.3390/ijerph192316321

**Published:** 2022-12-06

**Authors:** Ane Arregi, Aitana Lertxundi, Oscar Vegas, Gonzalo García-Baquero, Jesus Ibarluzea, Asier Anabitarte, Ziortza Barroeta, Alba Jimeno-Romero, Mikel Subiza-Pérez, Nerea Lertxundi

**Affiliations:** 1Faculty of Psychology, University of the Basque Country (UPV/EHU), 20008 San Sebastian, Spain; 2Environmental Epidemiology and Child Development Group, Biodonostia Health Research Institute, Paseo Doctor Begiristain s/n, 20014 San Sebastian, Spain; 3Spanish Consortium for Research on Epidemiology and Public Health (CIBERESP), Instituto de Salud Carlos III, C/Monforte de Lemos 3-5, 28029 Madrid, Spain; 4Department of Preventive Medicine and Public Health, Faculty of Medicine, University of the Basque Country (UPV/EHU), 48940 Leioa, Spain; 5Ministry of Health of the Basque Government, Sub-Directorate for Public Health and Addictions of Gipuzkoa, 20013 San Sebastian, Spain; 6Bradford Institute for Health Research, Bradford BD9 6RJ, UK

**Keywords:** environmental noise, children, sleep habits, INMA, directed acyclic graphs, socioeconomic status

## Abstract

Environmental noise is considered the second most serious environmental risk factor in Europe. However, little evidence exists regarding its impact on health and sleep in children, and the results are inconclusive. In this study, we aim to analyse the effect of environmental noise exposure on 11-year-old children’s sleep habits. Data were collected from 377 participants in the INMA-Gipuzkoa (INfancia y Medio Ambiente) cohort project using both parent-reported and actigraphic sleep measures. The results revealed that 60% of children have a day-evening-night environmental noise exposure (L_den_) of above 55 dB, which is defined as a “high noise level”. No differences in noise exposure were observed between different socioeconomic groups. However, no effect of environmental noise was found on sleep variables. The paper highlights the importance of studying how environmental noise may affect children’s sleep.

## 1. Introduction

Environmental noise is defined as any sound derived from human activity that is unpleasant, unwanted or harmful [1]. It includes road, rail and air traffic noise, noise emitted by industrial activity, noise from other leisure activities, and noise emitted by personal electronic devices, with road traffic being the main source in urban environments [2]. Environmental noise is considered the second most serious environmental risk factor in Europe [3]. In order to quantify the level of exposure to environmental noise and its impact on health, the most commonly used indicator is day-evening-night environmental noise exposure (L_den_), which describes an average sound pressure level over all of the day, evening and night periods of a year, with 5 dB and 10 dB penalties for evening and night-time, respectively [4]. Noise levels above 55 dB L_den_ and 50 dB L_night_ are defined by the European Union’s Seventh Environment Action Programme (7th EAP) and Environmental Noise Directive (2002/49/EC) [5] as “high noise levels”. The European Environment Agency (EEA) estimates that 113 million citizens are suffering from exposure above recommended values in Europe. This means that at least 20% of the total population are exposed to high levels during the day-evening-night period and more than 15% are exposed to these levels during the night-time period [6].

High environmental noise exposure has mainly been associated with immediate health effects [7], such as auditory health impacts [8], psychological problems [9], physical effects [10], impaired learning and cognitive performance [11,12] and sleep disturbances [13,14]. Although less research has been conducted on chronic effects, noise impacts on the cardiovascular system have been extensively studied [15,16] and are considered likely to cause hypertension and heart disease [17]. The association between environmental noise exposure and metabolic diseases has also been studied, with the former being found to increase the risk of developing diabetes [18].

Most noise effects in the adult population are associated with sleep habits, sleep disturbances and sleep quality [2]. Children are considered to be particularly vulnerable to the effects of environmental noise, mainly because they have developed fewer coping strategies and childhood is a vulnerable period for brain maturation and cognitive development [19,20]. Sleep plays an important role in these processes [21] so this issue is a vital one in relation to this population. However, research into how environmental noise may affect sleep in children is limited and inconclusive. A cross-sectional study with 7-year-old children in Norway observed no association between road traffic noise and parental reports of their children’s sleep duration or sleep problems in the total study population, although a statistically significant association was observed among girls [22]. Another cross-sectional study carried out with 9–12-year-old children in Sweden found a significant exposure-effect relationship between road traffic noise and sleep quality in children, with exposure being associated with more sleepiness problems during the day, although not with problems of falling asleep at night [23]. Another study observed the effect of night-time noise on sleeping problems, particularly problems falling asleep. In this case, the association was only significant when exposure was measured on the least exposed façade of the child’s home, but not when measured on the most exposed façade. The authors explain this phenomenon by noting that bedrooms are usually adjacent to the quieter façade, so noise levels on that side of the house are a better reflection of the exposure children suffer at night-time [24].

Noise effects on children’s health are believed to derive from the same mechanism as in adults [21]; noise exposure stimulates the endocrine system and the autonomic nervous system, resulting in a higher concentration of stress hormones, increased oxidative stress and inflammation [15,25]. Catecholamine and cortisol secretions have been studied as chronic stress indicators in response to aircraft and road traffic noise. Studies in adults have observed an association between salivary cortisol levels and environmental noise exposure [26,27], whereas studies in children are fewer and have reported inconsistent results [28,29,30], probably due to the intense development and/or decline of stress-related brain regions (amygdala, hippocampus and prefrontal cortex) prior to adulthood [31].

The answer to the question “How does environmental noise affect sleep in children?” may involve other factors that impact environmental noise or sleep in children, as well as the interrelation between them. These factors can be classified into different groups in accordance with their nature: environmental factors, social factors, family factors and individual factors. Concerning environmental factors, urban planning and urban morphology may influence environmental noise [32]. Regarding social factors, being involved in bullying has been associated with sleep disturbances among school-aged children [33,34]. Also, a positive school environment is believed to reduce the prevalence of bullying [35], and school provides an ideal framework for promoting children’s mental health [36]. As it regards family factors, socioeconomic status (SES) has been associated with several outcomes in children, including environmental noise exposure differences due to the unequal urban distribution of different socioeconomic groups [37], sleep quality [38], stress levels, mental health, physical activity and the prevalence of bullying [39,40]. Finally, several individual factors have been associated with sleep in healthy school-aged children. For instance, gender differences have been observed in terms of sleep quality among adolescents [41], with girls usually reporting poorer sleep and more sleeping problems [42], and higher levels of cortisol have been associated with more sleep problems in children [43]. Moreover, psychological well-being [44] and physical activity [45] have been negatively associated with sleep disturbances, whereas smartphone usage [46] is positively associated with this same factor. As mentioned above, good sleep is of vital importance during childhood; during this period, sleep disturbances are associated with several poor health outcomes, such as metabolic and cardiovascular problems, alterations in cortisol levels, obesity and poorer mental health [47,48,49,50].

Given the effect of environmental noise on adults’ sleep and the vulnerability of children to environmental factors, studying noise exposure in children and its effect on sleep is of the utmost importance. In this study, our first aim was to describe environmental noise exposure among 11-year-old children from a cohort in northern Spain, and to explore the association between SES and noise exposure. Our second aim was to test the correspondence between parent-reported sleep and sleep as measured by an actigraph. Finally, the main aim of the paper is to explore the effect of noise exposure on children’s sleep habits, with the hypothesis that children living in noisier houses would have poorer sleep outcomes. We used Directed Acyclic Graphs (DAGs) in order to study the complex pathways proposed in the theoretical framework.

## 2. Materials and Methods

### 2.1. Study Population

The study population comprised children participating in the 11-year follow-up phase of the INMA-Gipuzkoa cohort project (www.proyectoinma.org (accessed on 1 December 2022); INfancia y Medio Ambiente). The main aim of the INMA project is to analyse the association between exposure to environmental factors and effects on children’s health and physical and neuropsychological development. The INMA project is a prospective cohort study comprising seven different study areas [48]. The present study uses data from the INMA-Gipuzkoa cohort because Gipuzkoa is the only autonomous community (region) in which there is a specific regulation for estimating noise [49]. Thanks to this regulation, municipalities with more than 10,000 inhabitants are obliged to compile a noise map, whereas according to the Environmental Noise Directive [50], this obligation only applies to municipalities with over 100,000 inhabitants. 

The Gipuzkoa cohort project started in 2006, when participants’ mothers were informed of the initiative and recruited during the first trimester of pregnancy at Zumarraga Hospital. The following inclusion criteria for mothers were established: being older than 16 years, having the intention of giving birth in their referral hospital, having a single pregnancy, not having followed an assisted reproduction programme and not having communication problems. Since recruitment, data have been collected during several follow-up phases. In this study, we used data from the 11-year follow-up, which yielded a sample of *n* = 377, 54.1% of which were girls and 45.9% boys with an average age of 10.8 years (SD = 0.2). A subset of 135 participants accepted to wear an actigraph for 7 consecutive days. All participants gave their written informed consent before enrolling in the study and subjects’ parents were asked to complete some questionnaires about their children. The procedure was approved by the ethics committees of the hospitals in the region involved. 

### 2.2. Environmental Noise Measurement

An estimate of day-evening-night noise (L_den_) exposure was obtained for each subject at their home, along with an equivalent estimate of noise exposure during the day (L_day_) at school, reflecting noise immission levels as measured on the buildings’ façades. L_evening_ and L_night_ in subjects’ homes were also measured. These levels were calculated following the measurement methods recommended in Decree 213/2012, Royal Decree 1513/2005 and Decision No 1386/2013/EU [51,52,53] of the European Parliament. Traffic noise emission levels were calculated using the SoundPLAN^®^ acoustic software package, which enables users to calculate the sound power per metre at the emission source. In industrial environments, emissions are calculated by taking measurements at points in which the predominating noise origin is industrial, in order to identify relevant sources. Sound emissions are then estimated in accordance with distance to the point of measurement. The data obtained from rail, road and industry are used to create façade maps. In the resulting map, total environmental noise exposure in the building façades at a height of 4 metres is represented; the immission noise level at the building façades is measured, taking into account emission and propagation of all the relevant noise sources. In this study, noise exposure estimates were obtained by calculating the mean of all projected points on the building in question. Evening and nocturnal environmental noise at home was used to analyse the effect of environmental noise on sleep, since this time period usually corresponds to children’s sleeping hours. Noise maps were also compiled for municipalities with between 6000 and 10,000 inhabitants, following the same methodology.

### 2.3. Sleep Measurement

Both parent-reported and actigraphic data were collected, since the combination of actigraphy and questionnaires about sleep habits is deemed an efficient method for estimating sleep time and efficiency [54]. Subjects’ parents were asked to complete a questionnaire about their children’s sleep habits during the last year (Appendix A). Questions on bedtime, estimated time required to fall asleep and time to wake up were used to estimate four sleep outcomes: time in bed (hours), sleep period (the period the subject is believed to sleep in hours), sleep latency (hours) and sleep efficiency (percentage), measured by asleep period divided by time in bed [55]. Furthermore, a subset of 135 participants agreed to wear an actigraph (GENEActiv) for 7 consecutive days. The actigraph was placed on their non-dominant wrist (day and night). No demographic differences were observed between those who wore the actigraph and those who did not. Those participants who wore an actigraph were required to complete a sleep diary every morning when they woke up, specifying their bedtime the night before and their wake up time in the morning. All data from the device were downloaded and processed using the R-package GGIR, with information obtained from the sleep diary being used to guide accelerometer-based detection in the event of these data being available [56]. Sleep measures were obtained for each day: time in bed (hours), nocturnal sleep (hours), sleep latency (hours), sleep efficiency (percentage) and diurnal rest. The plain average of all available data was used for each variable. 

### 2.4. Other Variables Assessed

#### 2.4.1. Socioeconomic Characteristics and Number of Stressful Family Events

Other variables were also assessed using information provided in the questionnaires completed by both subjects and their parents. Parents were asked to answer a set of questions designed to gather data on family characteristics, including mother’s age, education level and type of work. Social class was calculated on the basis of the mother’s occupation, and grouped into 5 levels, with lower levels indicating more wealth. Similarly, area-level SES was calculated for each subject using a census deprivation index based on the MEDEA2011 project; the variables used were percentage of manual workers, unemployment, percentage of temporary workers and insufficient instruction [57,58]. Parents were also asked about any stressful family events that may have occurred since the child’s birth: change of residence, change of school, parental separation, death of a relative and/or hospitalisation of a relative (Appendix A). The sum of the number of stressful events was used.

#### 2.4.2. Hair Cortisol Concentration

In order to measure hair cortisol levels, trained staff cut strands of hair from the posterior vertex area of the participants’ heads, following a guideline published by the Society of Hair Testing [59]. The first 3 cm of outgrowth were analysed, since this hair segment reflects hair growth over the 3 months prior to hair sampling [59]. Because hair grows at a rate of 1 cm per month, analysing hair cortisol concentration in each cm provides month-to-month approximates of systemic cortisol levels. All analyses were performed in the Clinical Chemistry Laboratory of the University of Linköping (Sweden). Hair cortisol was extracted and analysed using a competitive radioimmunoassay. The method is fully described elsewhere [60]. 

#### 2.4.3. BMI and Physical Activity

Anthropometric measures were collected at the children’s schools. BMI was calculated by dividing the children’s weight in kilograms by their height in metres squared (kg/m^2^). Physical activity was measured using two methods. First, data were obtained from the actigraph (GENEActiv) in the subset of participants asked to wear that device. Raw data from the device were downloaded and processed using the R-package GGIR to obtain the total minutes per day spent on each type of activity (moderate or vigorous), which were then summed to calculate an overall moderate or vigorous physical activity level (minutes/day). Second, information on physical activity was collected through a questionnaire (Appendix A). Trained interviewers administered a questionnaire to parents outlining 31 physical activities that children may engage in during a typical week, in and out of school hours, along with three sedentary activities (television/video, computer games/inactive games, and board games or other sedentary activities outside school). Parents were asked how many minutes their child spent on each activity on weekdays (Monday to Friday) and on weekends. The activities included in the questionnaire were classified as light (playing, sitting on swings, going to the theatre, etc.), moderate (walking, cycling, scootering, rollerblading, skating, etc.) or vigorous (swimming, baseball, football, basketball, etc.), depending on their calorie consumption (MET). Both, moderate and vigorous activities were measured in minutes/day and added up in order to calculate overall physical activity. 

#### 2.4.4. Questionnaires Completed by Children 

Children were also asked to complete some questionnaires. Smartphone (computer, tablet) usage was measured in this way, with children being asked about their use of electronic devices at bedtime and during the night, with a dichotomous variable (Yes/No) being created for this purpose. Personal skills and abilities were measured using the Spanish version of the Strengths and Difficulties Questionnaire (SDQ) [61,62]. This instrument comprises five subscales: ‘Emotional symptoms’, ‘Behaviour problems’, ‘Hyperactivity’, ‘Peer problems’ and ‘Prosocial behaviour’. The total score used in this study was the sum of the first four subscales, with higher scores indicating more behavioural problems. Children were also asked to complete the Kidscreen-27 questionnaire, in order to assess their health-related quality of life. The Spanish version of the questionnaire has been validated and found to have adequate psychometric properties [63]. The Kidscreen-27 [64] is a self-report questionnaire comprising 27 items divided into five subscales. Only the psychological well-being and school environment subscales were used in the present study, with higher scores indicating better health-related quality of life in these areas. Finally, subjects’ propensities to be victims and bullies/victims were also estimated. Children answered a short version of the Olweus Bully Victim Questionnaire (OBVQ) [65], which has been found to have adequate psychometric properties [66]. The questionnaire consists of a standardised definition of bullying and 16 questions; the first eight items are related to victimisation behaviours and the second eight to bullying behaviours. A dichotomised variable was created [67]; when participants responded ‘it happens 2 or 3 times a month’ or more often to at least one of the items, they were categorised as a victim or both a bully and a victim.

### 2.5. Data Analysis

The data were analysed using R software v.4.0.3 [68]. First, we studied the symmetry of the variables, applying transformations when needed in accordance with Tukey’s ladder of powers [69]. Descriptive statistics and comparisons between socioeconomic groups in terms of exposure to environmental noise were likewise computed by applying the most appropriate test in each case. Furthermore, we tested correlations between reported and actigraphic sleep variables in the 135 sub-sample data. The group wearing an actigraph and the group not wearing the device were also compared in terms of environmental noise exposure and reported sleep outcomes. 

A Directed Acyclic Graph (DAG) [70] was compiled following Tennant et al.’s [71] recommendations. Ideally, DAGs represent sets of hypothesised causal relationships [70,71,72]; specifically, the DAG reported here (Figure 1) represents our review-based assumptions regarding the process underlying sleep habits. To determine whether the first version of our DAG was technically valid, we used Textor et al.’s DAGitty software [72] to identify its testable implications. Testable implications are properties of joint distribution that are dictated by the model structure, based on the rules of d-separation. [73,74], More precisely, these implications are the pairwise marginal and conditional independencies implied by a DAG [75]. Conditional independencies were tested by applying the most appropriate test in each case. If not satisfied in the data, these constraints allowed us to reject or modify the model. As a result of this process, the first version of our DAG was modified to include formerly missing relationships between school environment and psychological well-being, between BMI and bullying, and between sex and physical activity. It is this modified version that is presented here (Figure 1).

Once a DAG has been created, the covariate adjustment, needed to answer a specific question, can be achieved by applying the backdoor criterion established and explained by Shpitser et al. [76]. This process determines the minimal and sufficient adjustment set of variables that block all non-causal paths between the exposure and outcome variable, without blocking causal paths. We used Textor et al.’s DAGitty software [72] to identify the set of adjustment variables required to test direct and indirect effects of environmental noise on sleep habits (the estimation of which was the main goal of our research) [74]. Area-level socioeconomic status was identified as a minimal sufficient adjustment variable for testing the total effect of environmental noise on sleep habits. Regarding the direct effect of environmental noise on sleep habits, the minimal sufficient adjustment set of variables were bullying, cortisol levels, physical activity, psychological well-being, SES, sex and number of stressful events. Note that DAGs and the backdoor criterion are compatible with both linear and nonparametric approaches, and adjustments were made in this study work by means of linear modelling (i.e., within the framework of the general linear model). Not standardized partial regression coefficients (β-estimates) have been reported to reflect the changes in the response variable when environmental noise was increased by one unit (dB), while other predictors remained constant.

## 3. Results

### 3.1. Environmental Noise Exposure and Socioeconomic Differences

Table 1 shows the descriptive statistics for the exposure and outcome variables considered. In terms of L_den_ at home, 60% of children had a day-evening-night environmental noise exposure of over 55 dB which is defined by the 7th EAP as a ‘high environmental noise level’. Moreover, 25% were exposed to over 50 dBA during the night (Figure 2). Table 2 shows environmental noise exposure in accordance with socioeconomic variables. When groups were compared, statistically significant differences were found only in the deprivation index. However, no clear tendency from lower to higher socioeconomic groups was observed. 

### 3.2. Correspondence between Sleep Measurement Methodologies

Table 1 shows the descriptive statistics of the four subjective sleep outcomes considered in this study, estimated using the questionnaires completed by the children’s parents. The results for the total sample indicate that parents think their children spend, on average, 10.22 h in bed (SD = 0.4), 9.37 of them sleeping (SD = 0.47). Consequently, the sleep latency value was 0.85 h and sleep efficiency was extremely high (91% on average). No differences were observed in the reported sleep variables between those wearing an actigraph and those not wearing it. The mean nocturnal sleep and sleep latency values in accordance with the categorical variables analysed in this study are shown in Appendix A (Appendix A). In the sub-sample in which both parent-reported and actigraphic data were collected, parent-reported sleep was similar to that reported for the total sample; children spend, on average, 10.23 h in bed (SD = 0.4), 9.4 (SD = 0.43) of them sleeping, and the sleep latency value was 0.84 h and sleep efficiency was 92% on average. According to the data obtained from the actigraph, children spend, on average, 9.1 h in bed (SD = 0.6), 7.7 of them sleeping (SD = 0.6). The sleep latency value was, therefore, 1.4 h and sleep efficiency was about 85%. Participants usually slept 0.9 h during the day (SD = 0.6).

Correlation among all sleep outcomes is reported in Appendix A. Correlations between reported and actigraphic sleep variables were non-significant and very small in all cases (Figure 3). Reported sleep time and efficiency estimation were higher than sleep as measured by actigraph, with all sleep values measured by actigraph being significantly lower than their parent-reported variable pair value, with the exception of sleep latency.

### 3.3. Testing the DAG Model

The code for the model created by DAGitty is given in Appendix A and the model itself is shown in Figure 1. Two models were created with the same skeleton: one suitable for actigraph measurements and the other for reported ones, since these measurements were different from each other. The 84 testable implications of the model generated from actigraphic data and the 81 testable implications of the subjective model, along with the values of the hypothesis test carried out, are outlined in Appendix A. The statistical tests used depended on the nature of the tested variables, with T-values, F-values, Kruskal-Wallis, Chi-sq and Welch’s T-values all being reported. Note that 4 out of the 165 tests were moderately statistically significant (0.04 < *p*-value < 0.05). Moreover, in all cases, the correlations observed were weak, meaning that, overall, we found insufficient reason to justify including new relationships in the DAGs. 

### 3.4. The Effect of Environmental Noise on Sleep Habits at 11 Years of Age

A suitable DAG model was created to explore the effect of environmental noise on sleep habits (Figure 1), as proven by the testable implications. The deprivation index that was used as a proxy variable for socioeconomic status was established as a minimal sufficient adjustment variable to test the total effect of environmental noise on sleep habits. Marginally significant (0.05 < *p* < 0.1) effects of nocturnal environmental noise were observed on time in bed, sleep latency and reported sleep period. However, these effects did not follow the same trend, since the effect on time in bed and sleep latency were negative, whereas the effects on reported sleep period were positive. Consequently, no evidence was found that evening or nocturnal environmental noise had a significant total effect on sleep habits (Table 3). The minimal sufficient adjustment set of variables used to test the direct effect of environmental noise on sleep were bullying, cortisol levels, physical activity, psychological well-being, SES, sex and number of stressful events. Regarding the direct effect of environmental noise on sleep, only marginally significant effects were observed (0.05 < *p* < 0.1), and no clear trend was identified (Table 4). Adjusting the model in accordance with propensity to be a bully/victim and SDQ (instead of victim propensity and psychological well-being as measured by the Kidscreen 27) returned similar results (Appendix A).

## 4. Discussion

As mentioned earlier, although children are considered particularly vulnerable to the effects of noise, the number of studies focusing on the impact of environmental noise on children’s health is limited. Due to the high degree of urbanisation and our current lifestyle, noise has become an ever-present factor. According to the European Environment Agency [3], environmental noise is considered the second most serious environmental risk factor in Europe, and several detrimental health effects have been described [10]. 

Regarding the description of environmental noise exposure, the results of our study revealed that 60% of children are exposed to a day-evening-night environmental noise level of over 55 dB, and 25% are exposed to over 50 dBA during the night. These figures are comparable to those pertaining to larger cities less than 100 km from the study area, such as Bilbao or San Sebastian, in which 61% and 58% of the population are exposed to high levels of noise, respectively [6]. It is interesting to note that exposure levels in a less urbanised area, with small towns of between 6000 and 15,000 inhabitants, are comparable to those reported for larger cities such as Rome (58% above 55 dB L_den_), and are even higher than in certain capital cities such as Lisbon (44.5%) and Madrid (38%) [6].

No socioeconomic differences in noise exposure were observed in the present study. Few studies have explored social differences in environmental noise exposure, and those that have offer low comparability due to methodological differences [37]. Some of these studies reported higher environmental noise exposure among groups with a lower socioeconomic status, thereby indicating an increased risk for the most vulnerable groups [77,78,79]. However, other studies observed higher exposure levels in higher socioeconomic status groups [78,80,81]. A systematic analysis of data from European Union Statistics on Income and Living Conditions (EU-SILC) publications concluded that self-reported noise annoyance varies between countries; in the most north-western European countries, a higher noise exposure prevalence was observed among individuals living below the relative poverty level, whereas no differences or a lower prevalence was observed in low-income groups in south-eastern countries [82]. However, noise annoyance is a self-reported measurement and does not necessarily reflect real noise exposure. In the present study, the low variability across the sample may explain why we did not observe any clear trend in terms of noise exposure and the socioeconomic groups identified. It is important to highlight the fact that the study area may be considered a semiurban area in which the smallest town has 6007 inhabitants and the largest one 15,191 [83]. The fact that it is a semiurban area and not a large city may explain the homogeneity of the sample and consequently the lack of socioeconomic differences in relation to noise exposure. Moreover, the deprivation index, the variable used to represent area-level socioeconomic status in the present study, was calculated in 2011 and may therefore be outdated. Further research into the social distribution of environmental noise exposure is required, since estimating social inequalities would allow us to design evidence-based preventive measures to protect against this potential problem.

In relation to our second aim, we analysed both children’s sleep and the correspondence between sleep estimated by parent reports and sleep measured by actigraph. The National Sleep Foundation recommends that, at age 11, children sleep between 9 and 11 h a night [84]. Our results regarding subjectively estimated sleep duration were in line with this recommendation, whereas sleep duration measured by actigraph was lower in almost all cases. This indicates that sleep estimations using questionnaires completed by parents are higher than those measured by actigraph. Our results suggest that parents think their children sleep more than actigraph measures actually show, with very little correlation between them. This is consistent with that reported by other studies that found poor correlation between sleep questionnaires completed by parents and actigraphic data, with parents tending to overestimate the time for which their child sleeps. Specifically, parents usually report earlier bedtimes and later wake up times in comparison to actigraphic measures [85,86]. Parents’ overestimation of sleep duration varies in the literature from 30 to 113 min per night, and difficulty calculating sleep latency may be the main reason for this. These values are consistent with our results, which revealed that reported sleep duration was, on average, 100 min longer and reported sleep latency 33 min lower. Moreover, the questionnaire on sleep habits completed by parents was based on sleep habits over the past year, and agreement rates between the questionnaire and actigraphic data may be lower the less recent the period covered by the questionnaire. In other words, answers have been influenced by memories, experiences, and expectations (recall bias) [54]. Both methods have their limitations: questionnaires are based on parents’ ability to assess their children’s sleep, whereas wrist actigraphic data may not distinguish between a person who is asleep and one who is awake but lying still, or between a sleeper with high motility and someone who is awake. Sleep recording with an actigraph also involves methodological difficulties, such as the monetary cost of the devices and the need to wear them during seven consecutive days. Consequently, obtaining large samples of actigraphic records is difficult. Since statistical power depends on sample size, said power was lower for the sample in which sleep was measured using actigraph (*n* = 135) than in the sample in which sleep was estimated subjectively (*n* = 377). For example, when using the first sample, we were able to detect associations with R^2^ ≥ c.30%, whereas when using the second sample, we are able to detect associations with R^2^ ≥ c.10%.

The final aim of the present study was to test the effect of environmental noise on children’s sleep habits. As shown in Figure 1, environmental noise may affect sleep habits both directly and indirectly by altering cortisol levels. However, no significant total or direct effect was observed, meaning that we cannot conclude that environmental noise affects sleep habits through either pathway. In relation to actigraphy-based sleep estimations, no effect of evening or nocturnal environmental noise was observed on any sleep outcome, nor was any clear trend found. This is consistent with the only other study to have assessed sleep using wrist-actigraphy, which observed no exposure-response association between road traffic noise and sleep variables measured using this method [23]. In relation to subjective sleep estimation, a trend was observed, although it was non-significant in all cases: both evening and nocturnal environmental noise levels were found to have a positive effect on reported time in bed, sleep period and sleep efficiency, and a negative effect on sleep latency. This contradicts our hypothesis that children living in noisier houses would have poorer sleep outcomes. Previous studies on traffic noise and sleep in children have found an association between noise and some sleep outcomes. For example, Tiesler et al. [24] found that night noise on the least exposed façade was associated with sleeping problems, particularly problems falling asleep. It is important to note that, in that study, three dichotomous variables were used to measure the presence of sleep problems (specifically difficulties falling asleep or difficulties sleeping through the night), whereas in our study, sleep latency was measured by calculating the difference between total time in bed and nocturnal sleep. Öhrström et al. [23] found a relationship between traffic noise and sleep quality, as well as problems with daytime sleepiness, but not with problems falling asleep. In this case, sleep quality was measured by asking ‘*How well do you usually sleep*?’, whereas in our case, sleep was assessed by sleeping time and sleep efficiency (ratio between nocturnal sleep and total time in bed). In the study by Weyde et al. [22], sleep duration was assessed by asking mothers: ‘How many hours of sleep per night does your child usually obtain on weekdays?’ Consistent with the results of our study, no association was found between road traffic noise and sleep duration in the total study population. However, when population was stratified by sex, a statistically significant association was found among girls, with a reduction in sleep duration being observed in those experiencing higher road traffic noise levels. It is important to highlight the fact that only this last study used both evening and night-time environmental noise, merged into the L_en_ parameter [22], whereas the other two studies mentioned above used the L_den_ and L_night_ indicators [23,24]. Given that children usually go to bed before 11 p.m., it is important to take evening environmental noise into account. The small number of studies focusing on the issue, coupled with the methodological differences that exist between them, makes it difficult to draw clear conclusions.

One possible reason for our failure to observe any effect of noise exposure on sleep in children may be that children are less likely to wake up once they have fallen asleep. For example, one study concluded that children required a traffic noise level of 10 dB(A) or higher in order to be awakened [87]. It therefore seems that children are more protected against noise-related sleep disruptions than adults. However, although we did not find sleep duration, sleep latency or sleep efficiency to be affected by noise, other reactions may occur. Studies on adult sleep have concluded that noise can disturb sleep by changing sleep stages, even though this does not necessarily disturb the macrostructure of sleep [88]. Cardiac responses to noise, such as higher heart rate, may also occur in response to environmental noise. Moreover, it seems habituation to traffic noise does not occur in the case of cardiac responses [89]. Consequently, although no effect of environmental noise was observed on the sleep outcomes considered in this study, this should not lead to the conclusion that children are not disturbed at all by noise during sleep. Moreover, it is possible that other sleep disruptors exist that we were unable to detect using the methods employed in the study.

The impact of noise on sleep in children may also vary in accordance with the source of noise. A study from South Korea found that road traffic noise was not associated with sleep disturbances, whereas aircraft noise did significantly alter sleep in children [90], despite being lower than road traffic noise. Therefore, noise characteristics rather than just the total volume of noise may also be an important factor. Road traffic noise is a continuous noise, while aircraft noise is highly intermittent and characterised by single dominant events. Foraster et al. [12] have highlighted the relevance of noise characteristics other than average noise levels, such as noise fluctuation, for example, in the effects of noise on children’s health and neurodevelopment.

Another possible reason why we did not observe any effects may be that noise levels were calculated outdoors and were estimated by calculating the mean of all projected points on the subject’s building façade at a height of 4 metres, leading to possible misestimates of noise exposure inside the house. Information regarding where the children’s bedrooms were located was not considered, and nor were participants asked if their façades were insulated, whether they lived on the lower or upper floors (below/above a height of 4 m), slept in rooms not facing the street, slept with opened or closed windows, etc. Consequently, the average noise level on the building’s façade may not reflect noise exposure in the child’s bedroom. Moreover, although window glazing was also taken into account, since most of the subjects had double glazing, this variable did not provide us any extra information. Other studies have also pointed out this problem [22,24], and indeed, one reported an effect of noise levels on the least exposed façade, but not on the most exposed one [24]. As mentioned earlier, the fact that bedrooms are usually located in the most silent part of the house may explain this observation. Including more information about the location of children’s bedrooms and indoor noise exposure should help us gain greater insight into this effect.

Moreover, the homogeneity of the sample may also explain the reason why no effects were observed. As explained earlier, low variability was a feature of the sample, meaning that sleep variables did not vary greatly and there was no group of children with sleeping problems. In order to gain a better understanding of the situation, further research is required with more heterogeneous samples and greater variability. Also, subjective sleep was estimated by parents, not the children themselves. Sleep duration as estimated by children themselves may be more accurate, as parents may know when their children enter or leave their bedroom, but not usually how they spend the time that transpires between these two time points. It is important to note that a good night’s sleep does not only imply falling asleep and sleeping for a certain number of hours, but also involves other features, such as not waking up during the night, self-satisfaction of sleep and feeling rested during the day. Other sleep variables, such as sleep microstructure and heart rate, may be affected by high environmental noise levels [88,89]. Consequently, other responses should be analysed in future studies, in order to gain a deeper understanding of the issue.

Despite these limitations, the study has some strengths that are worth highlighting, the main one being the fact that it was carried out with 11-year-old children. Given the scarce amount of evidence that currently exists in this field, studies exploring how noise affects children are of the utmost importance. Moreover, a complex DAG model was created with a large number of variables, with the aim of obtaining a more complex and realistic image of how environmental noise may affect sleep habits among children. As tested by means of testable implications (conditional independencies), this DAG model is consistent with our data, suggesting that it is valid for explaining the effect of environmental noise on sleep habits. Estimating causal effects is one of the most difficult tasks in applied health research, as many sources of bias are present in observational data, including confounding bias, endogenous selection bias and information bias [91]. DAG models allow a minimal sufficient set of variables to be identified, in order to obtain unbiased effect estimates [92] and have consequently become an interesting tool in epidemiological research. However, this is a cross-sectional study, and we cannot therefore assume causality; exposure may not be constant over time, hence the need for longitudinal studies and causal models to study causal effects. Moreover, social differences in environmental noise exposure were also explored in this study. More and more studies are focusing on social inequalities and their effects on health, but further research is still required into the social differences observed in environmental noise exposure, since identifying more vulnerable populations may contribute to the design of more effective interventions. It is also worth noting that, in this study, a more complex definition of environmental noise was used, with road and train traffic and industrial activity being considered, rather than just traffic noise, as is the case in most other studies [22,23,24]. Given that the children participating in our study lived in a not particularly densely populated, yet highly industrial area, with a railway line that passes through most of the villages, taking noise sources other than traffic into consideration was of major importance. The Environmental Noise Directive 2002/49/EC [49] on the assessment and management of environmental noise obliges Member States to compile strategic noise maps for all territories with a population of over 100,000, with L_den_ and L_night_ being the indicators used to assess health effects. However, the Autonomous Community of the Basque Country, where the study area is situated, obliges all local councils with a population of over 10,000 inhabitants to compile a strategic noise map [48] to enable the study of noise effects in less urbanised areas. Finally, we also considered both actigraph-based and parent-reported sleep, an important issue in light of the inconsistency detected between the two methods. Several studies have compared the two methodologies, finding that parents usually overestimate their children’s sleep [54,85,86]. However, of the studies mentioned above which analysed the effect of environmental noise on children’s sleep, only one included an actigraph-based measure of sleep [23].

As explained earlier, the fact that we did not observe any association between evening and nocturnal environmental noise and sleep duration, sleep latency and sleep efficiency does not mean that environmental noise does not affect sleep itself. Continuing with this idea and bearing in mind that children are considered to be a particularly vulnerable group [21], other noise effects should be studied in the future. It would also be interesting to include perceptions of environmental noise, since subjective noise sensitivity may affect the noise-response relationship. This paper highlights the importance of studying how environmental noise may affect children’s sleep, but also underscores the need to explore other responses, including that of the cardiovascular system and metabolic functioning, stress and immune responses, neuropsychological development, psychological well-being and other physical and emotional health factors.

## 5. Conclusions

The present study highlights the importance of studying the effects of environmental noise on vulnerable groups, such as children. The results reveal that the children participating in the INMA-Gipuzkoa cohort project experience high environmental noise exposure, comparable to noise levels in big cities. However, no effect of noise on sleep was observed. These results highlight the need for further research into this topic. Future studies should include indoor noise measures as well as environmental noise perception. Moreover, other responses such as noise annoyance, stress or immune biomarkers or neurodevelopmental outcomes should be included also, in order to obtain a holistic understanding of how environmental noise may affect children’s health.

## Figures and Tables

**Figure 1 ijerph-19-16321-f001:**
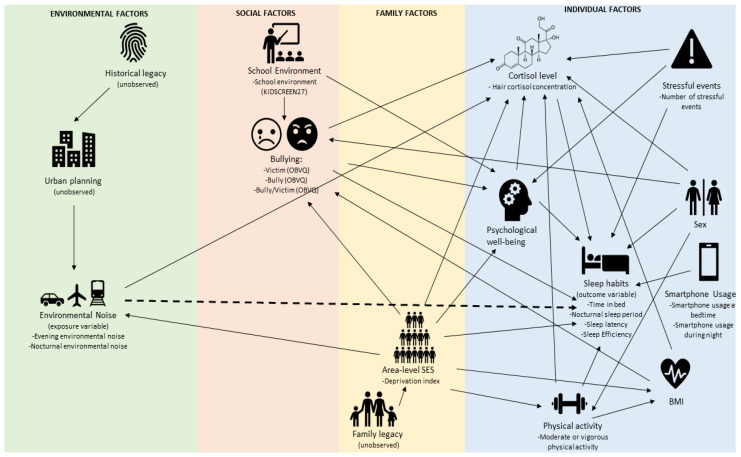
Directed Acyclic Graph (DAG) explaining the relationship between environmental noise and sleep habits in children. This model was used in the present study to test the effect of noise on children’s sleep (marked in dotted lines). Four sections are shown in the image in order to represent multiple levels of influence (ecological model); interactions between environmental, social, family and individual factors can alter children’s sleep habits (outcome variable). The variables and measurement scales used in each node are shown under the construct name. Environmental noise was considered an exposure variable in this study, and historical legacy, urban planning and family legacy were unobserved.

**Figure 2 ijerph-19-16321-f002:**
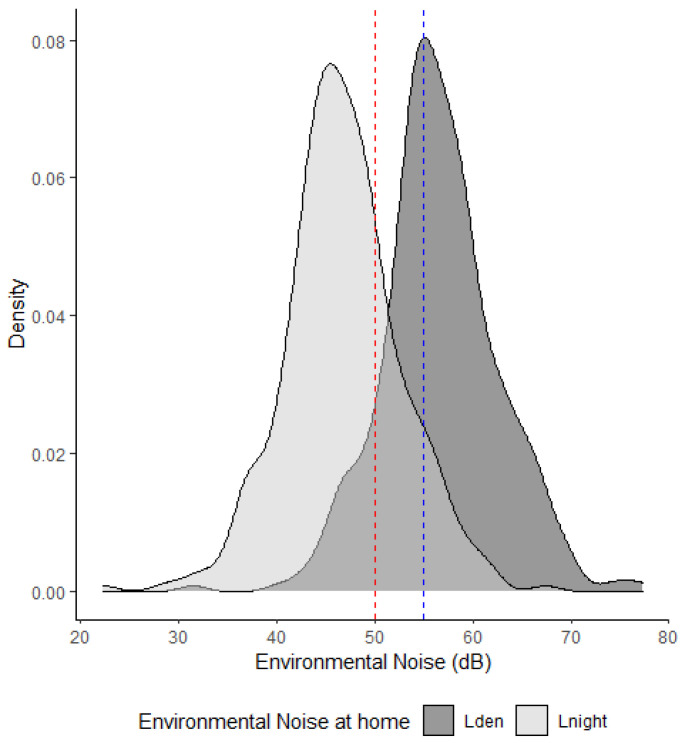
Environmental noise exposure density plot in the total sample: L_den_ and L_night_ at home. The dotted lines show high environmental noises defined by the 7th EAP as high noise levels.

**Figure 3 ijerph-19-16321-f003:**
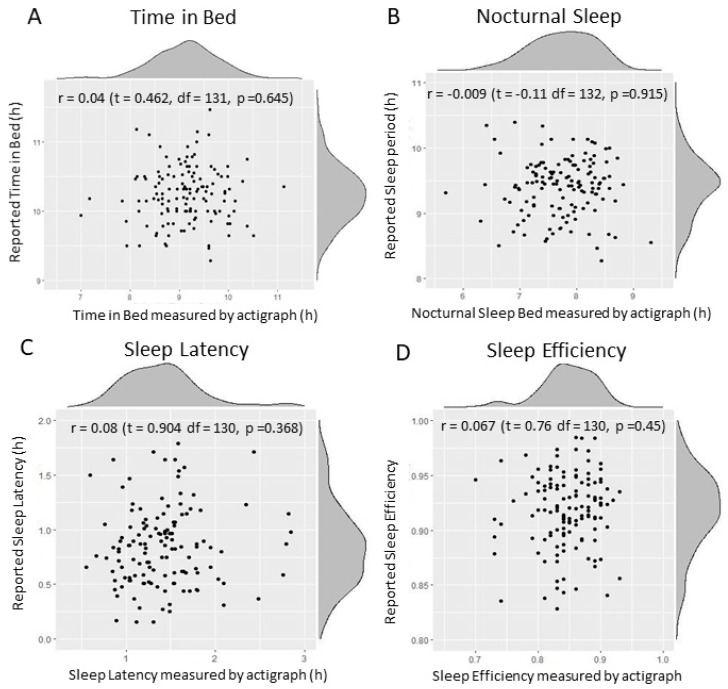
Scatter plots of sleep variables measured by actigraph and by questionnaires. Actigraphic measurements are shown on the x-axis, whereas the y-axis represents subjective measurements. Density plots for each variable and correlation estimates are also shown, with no significant correlations being observed.

**Table 1 ijerph-19-16321-t001:** Description and distribution of environmental exposure and response variables. Sample size (N), mean, standard deviation (SD), minimum and maximum values and quartiles are given for each variable.

Variable	N	Mean	SD	Min	Q1	Median	Q3	Max	IQR
*L_den_ at home (dB)*	328	56.7	5.9	31.5	53.3	56.2	60.0	77.4	6.7
*Evening environmental noise at home (dBA)*	328	53.7	6.2	28.8	50.2	53.4	56.8	80.1	6.6
*Nocturnal environmental noise at home (dBA)*	328	46.7	5.9	22.3	43.2	46.6	50.0	67.4	6.8
*L_day_ at school (dBA)*	323	49.8	6.1	39.9	44.7	52.0	54.9	60.3	10.2
*Time in bed (hours)*	135	9.13	0.64	7	8.72	9.2	9.55	11.1	0.83
*(Total) nocturnal sleep (hours)*	135	7.70	0.60	5.70	7.36	7.76	8.21	9.30	0.85
*Sleep onset latency (hours)*	135	1.40	0.44	0.55	1.08	1.40	1.61	2.90	0.53
*Sleep efficiency (%)*	135	85	4	70	83	85	88	93	5
*Diurnal rest (hours)*	135	0.93	0.58	0.00	0.54	0.84	1.15	3.20	0.61
*Reported time in bed (hours)*	369	10.22	0.41	9.14	9.96	10.20	10.47	12.48	0.51
*Reported sleep period (hours)*	371	9.37	0.47	7.93	9.07	9.43	9.65	11.67	1.72
*Reported sleep onset latency (hours)*	366	0.85	0.40	0.00	0.58	0.82	1.07	2.42	0.49
*Reported sleep efficiency (%)*	336	91	3	76	89	92	94	100	5

**Table 2 ijerph-19-16321-t002:** Exposure means for each socioeconomic group. The results presented in the table are based on the total sample. For each group, sample size (N), frequencies and environmental noise means are given.

		N	%	L_den_ at Home (dB) (SD)	Evening Environmental Noise (dBA) (SD)	Nocturnal Environmental Noise (dBA) (SD)	L_day_ at School (dBA) (SD)
	Overall mean exposure, by groups						
Neighbourhood SES (indicated by deprivation index)	Very High	25	6.63	57.3 (5.5)	54.1 (5.7)	47.8 (5.4)	46.1 (6.2)
	High	145	38.46	57.7 (6.1)	54.6 (6.2)	48.5 * (6.4)	50.2 (6.1)
	Medium	121	32.10	56. 1 (5.5)	53.6 (6.2)	45.1 (4.9)	49.9 (5.8)
	Low	70	18.57	55.0 (5.7)	51.6 (5.7)	45.5(5.7)	50.7 (6.2)
	Very Low	16	4.24	60.8 * (6.0)	57.3 (6.3)	50.7 * (5.9)	46.8 (5.6)
Social class: indicated by mother’s occupation	Very High	66	17.51	57.8 (5.9)	54.9 (6.5)	47.5 (5.6)	48.9 (6.2)
	High	60	15.92	57.0 (6.5)	54.2 (7.0)	46.9 (5.9)	50.7 (6.0)
	Medium	115	30.50	56.8 (5.9)	53.6 (6.2)	47.2 (6.0)	49.6 (6.2)
	Low	113	29.97	55.9 (5.1)	53.0 (5.1)	45.9 (5.6)	49.8 (6.2)
	Very Low	23	6.10	56.3 (8.2)	52.9 (8.1)	46.6 (8.1)	50.9 (5.0)
Mother’s education level	Primary	37	9.87	55.6 (5.1)	52.8 (5.1)	45.7 (5.6)	50.3 (6.1)
	Secondary	143	38.13	56.6 (5.3)	53.5 (5.4)	46.7 (5.7)	49.7 (6.2)
	University	195	52.00	57.0 (6.5)	54.0 (6.9)	46.9 (6.2)	49.8 (6.1)
Mother’s type of work	Manual	135	36.10	55.9 (5.7)	52.9 (5.7)	45.9 (6.0)	49.9 (5.9)
	No Manual	239	63.90	57.1 (6.0)	54.1 (6.4)	47.2 (5.9)	49.7 (6.2)
School type	Public	182	49.59	57.1 (6.3)	54.3 (6.8)	46.7 (6.6)	49.5 (6.25)
	Private	185	50.41	56.4 (5.6)	53.3 (5.7)	46.8 (5.7)	50.1 (5.9)

Note: (*) *p* < 0.05.

**Table 3 ijerph-19-16321-t003:** Total effects of evening and nocturnal environmental noise on sleep habits. Non-standardised B-estimates and 95% confidence intervals (in parenthesis) are shown.

		Evening Environmental Noise		Nocturnal Environmental Noise	
Measured by actigraph	Time in Bed	−0.005	(−0.023, 0.013)	−0.020	(−0.040, 0.002)
	Nocturnal Sleep	0.003	(−0.014, 0.020)	−0.007	(−0.026, 0.011)
	Sleep Latency	−0.008	(−0.020, 0.003)	−0.011	(−0.020, 0.001)
	Sleep Efficiency	0.080	(−0.040, 0.200)	0.090	(−0.030, 0.200)
	Dirunal Rest	−0.008	(−0.025, 0.009)	−0.001	(−0.018, 0.017)
Parent-reported	Reported Time in Bed	0.005	(−0.002, 0.012)	0.005	(−0.001, 0.013)
	Reported Sleep Period	0.007	(−0.002, 0.015)	0.007	(−0.001, 0.016)
	Reported Sleep Latency	−0.003	(−0.009, 0.004)	−0.002	(−0.009, 0.005)
	Reported Sleep Efficiency	0.020	(−0.030, 0.100)	0.030	(−0.050, 0.900)

Note: Model was adjusted for deprivation index.

**Table 4 ijerph-19-16321-t004:** Direct effects of evening and nocturnal environmental noise on sleep habits. Non-standardised B-estimates and 95% confidence intervals (in parenthesis) are shown.

		Evening Environmental Noise		Nocturnal Environmental Noise	
Measured by actigraph	Time in Bed	−0.004	(−0.023, 0.016)	−0.017	(−0.038, 0.017)
	Nocturnal Sleep	−0.001	(−0.019, 0.017)	−0.110	(−0.030, 0.009)
	Sleep Latency	−0.003	(−0.015, 0.010)	−0.006	(−0.019, 0.008)
	Sleep Efficiency	0.040	(−0.900, 0.200)	0.030	(−0.100, 0.170)
	Dirunal Rest	−0.008	(−0.029, 0.011)	−0.001	(−0.020, 0.021)
Parent-reported	Reported Time in Bed	0.006	(−0.020, 0.013)	0.005	(−0.003, 0.010)
	Reported Sleep Period	0.008	(−0.001, 0.017)	0.007	(−0.002, 0.017)
	Reported Sleep Latency	−0.002	(−0.009, 0.005)	−0.002	(−0.010, 0.006)
	Reported Sleep Efficiency	0.030	(−0.050, 0.100)	0.020	(−0.040, 0.100)

Note: Model was adjusted for victim propensity (bullying), hair cortisol level, physical activity measured by actigraph, psychological well-being measured by the Kidscreen 27, SES, sex and number of stressful events. Adjusting for bully/victim propensity and SDQ (instead of victim propensity and psychological well-being as measured by the Kidscreen 27) returned similar results.

## Data Availability

Not applicable.

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
