# Peer review of "Environmental Noise Exposure and Sleep Habits among Children in a Cohort from Northern Spain"

_ijerph, 2022, doi:10.3390/ijerph192316321_

Round 1

Reviewer 1 Report (Previous Reviewer 1)

Please state the reasons and scientific evidence for summing the sound energy from different sources. This is not a trivial question as they may differ in time and characteristic and it is not obvious from a response perspective to sum the energy.

In the description of the results referring to table 3 and table 4, can the results be more clearly described. When for example reading the text "Regarding the direct effect of environmental noise on sleep, only marginally significant effects were observed, and no clear trend was identified (Tables 3 and 4).

Please be more specific, for example which results would you consider marginally significant and which trends did you expect?

I am happy to see that the discussion brings up several important aspects, and also many interesting observations about sound characteristics and sleep quality aspects.

However I would be careful with wordings like: "and more and more detrimental effects have been described over recent years" Does the reference given actually support this statement, for the child population? I can not see that it does.

I would also refrain from using the wording: Objective sleep recordings for wrist actigraphy, they are merely detecting movements, that is interpreted by an algorithm as sleep or not. Please change.

This occurs in many places see i.e row 619, it is there impossible to work out what is meant if the wording is used in a not precise way.

I would also be more careful repeating what others have written such as "One reason for not observing any effect of noise exposure on sleep in children may be that children are less likely to wake up once they have fallen asleep, as they are about 10dB less sensitive than adults to awakening reactions and sleep disturbances" I guess that the authors are aware of the studies that have investigated sleep among children include a very small. For example the study referred to included 8 and 5 (maybe the same children in both experiments, that is not clear from the article). Maybe a slightly more cautious writing are at hand?

page 15, there are two e missing: slp microstructure

The conclusion mentiones: "other responses also" what is meant with this, it is a rather unprecise comment to make in a conclusion. could it be rephrased?

Author Response

REVIEWER 1

Please see the attachment, whre manuscript  with suggested changes is attached

Question 1. Please state the reasons and scientific evidence for summing the sound energy from different sources. This is not a trivial question as they may differ in time and characteristic and it is not obvious from a response perspective to sum the energy.

Answer 1. Environmental noise levels were calculated following the measurement methods dictated by EU policy and policy from the study area region (Decree 213/2012, Royal Decree 1513/2005 and Decision No 1386/2013/EU). The main difference between both regulations is that the regional decree (Decree 213/2012) states that all municipalities with more than 10,000 inhabitants must have a noise map, while in the European regulation  the obligation only applies to municipalities with over 100,000 inhabitants. The noise maps from smaller municipalities in the study area were created following same methodology.

Due to the study area characteristics (not particularly highly populated yet highly industrial area, with a railway line that passes through most of the villages) taking into account all these noise sources was of major importance. Therefore, the total noise exposure (or the combined source noise exposure) was used: road traffic noise, rail traffic noise and industry noise were considered. To measure this total noise, the noise emission and propagation of each of the previously mentioned sources is calculated. Then, noise level  immision in the building façade at a height of 4 metres is measured, taking into account all noise sources affecting. This information have been added in the methodology section, to obtain a more clear explanation about environmental noise measurement.

Below, another study using also several environmental noise sources is presented:

Essers, E.; Pérez-Crespo, L.; Foraster, M.; Ambrós, A.; Tiemeier, H.; Guxens, M. Environmental Noise Exposure and Emotional, Aggressive, and Attention-Deficit/Hyperactivity Disorder-Related Symptoms in Children from Two European Birth Cohorts. Environ. Int. 2022, 158, doi:10.1016/j.envint.2021.106946.

Decree 213/2012 of 16 October 2012 relating to the acoustic noise pollution in the Basque Autonomous Community, Boletín Oficial del País Vasco, 222, of 16 November 2012. https://www.legegunea.euskadi.eus/eli/es-pv/d/2012/10/16/213/dof/spa/html/webleg00-contfich/es/

Directive 2002/49/EC of the European Parliament and of the Council of 25 June 2002 relating to the assessment and management of environmental noise - Declaration by the Commission in the Conciliation Committee on the Directive relating to the assessment and management of environmental noise, Official Journal of the European Union, 18th July 2002. http://data.europa.eu/eli/dir/2002/49/oj

Royal Decree 1513/2015 of 16 December 2005, implementing 37/2003 law of November 17, 2003, on Noise, in relation to the evaluation and management of environmental noise, Boletín Oficial del Estado, 301, of 17 December 2005.  https://www.boe.es/eli/es/rd/2005/12/16/1513/con

Question 2. In the description of the results referring to table 3 and table 4, can the results be more clearly described. When for example reading the text "Regarding the direct effect of environmental noise on sleep, only marginally significant effects were observed, and no clear trend was identified (Tables 3 and 4). Please be more specific, for example which results would you consider marginally significant and which trends did you expect?

Answer 2. Results section has been edited in text, in order to describe them more clearly.

Question 3. I am happy to see that the discussion brings up several important aspects, and also many interesting observations about sound characteristics and sleep quality aspects. However I would be careful with wordings like: "and more and more detrimental effects have been described over recent years" Does the reference given actually support this statement, for the child population? I can not see that it does.

Answer 3.  Rephrased in text

Question 4. I would also refrain from using the wording: Objective sleep recordings for wrist actigraphy, they are merely detecting movements, that is interpreted by an algorithm as sleep or not. Please change. This occurs in many places see i.e row 619, it is there impossible to work out what is meant if the wording is used in a not precise way.

Answer 4. Wording objective sleep recordings have been rechecked in text and the use of “sleep measured by actigraph” has been selected.

Question 5. I would also be more careful repeating what others have written such as "One reason for not observing any effect of noise exposure on sleep in children may be that children are less likely to wake up once they have fallen asleep, as they are about 10dB less sensitive than adults to awakening reactions and sleep disturbances" I guess that the authors are aware of the studies that have investigated sleep among children include a very small. For example the study referred to included 8 and 5 (maybe the same children in both experiments, that is not clear from the article). Maybe a slightly more cautious writing are at hand?

Answer 5. Rephrased in text.

Question 6. page 15, there are two e missing: slp microstructure

Answer 6. Rechecked in text.

Question 7. The conclusion mentiones: "other responses also" what is meant with this, it is a rather unprecise comment to make in a conclusion. could it be rephrased?

Answer 7. Rephrased in text.

Reviewer 2 Report (Previous Reviewer 2)

Researchers responded appropriately to earlier questions about the study and believe it is now ready for publication. 

Author Response

REVIEWER 2

Researchers responded appropriately to earlier questions about the study and believe it is now ready for publication.

We would like to thank you for revising the manuscript

Reviewer 3 Report (New Reviewer)

ID: ijerph-2031333

Title: Environmental noise exposure and sleep habits among children in the INMA-Gipuzkoa cohort

Comment: major revision.

Detailed information:

Title: Writing an abbreviation in the title may be confusing.

Abstract

1) The Abstract contains too much redundancy and you need to refine the main outcome of the study.

2) Line 21-23, page 1: Hard to read and understand.

3) Line 30-32, page 1: Why did you write a Discussion-like sentence in the abstract? I suggest removing it.

4) Line 34, page 1: Writing an abbreviation in the title may be confusing.

1. Introduction

Line 49, page 2: I believe double quotations should be used instead of single quotations.

Line 54-112, pages 2-3: The logic of these paragraphs is confusing and vague. What do you really want to express? Which is the core scientific problem that needed to be addressed? I understand “Other factors” you wrote here is to coordinate with your GAP analysis yet this is not ideal. Writing a skeleton for your introduction before revising might help.

Line 63-66, page 2: Are children participants to the same high noise level standards as adults?

2. Materials and Methods

2.1. Study population

Line 138-141, page 3: The inclusion criteria need to be clarified. Reading the “being older than 16 years” before the “the 11-year follow up” is dizzy.

Line 141-142, page 3: Why did you choose the 11-year follow up? Any reason for it?

2.2. Environmental noise measurement

Line 160 and 162, page 4: What’s “façades”? Do you mean “facades”, the surface? If yes, what does this stand for?

  2.3. Sleep measurement

Line 171-176, page 4: Is this a validated questionnaire? How do you ensure its validity and reliability?

  2.4. Other variables assessed

Line 193, page 4: Why the font style is different here? Ensure no format issues before submitting.

Line 189-248, pages 4-5: 1) Simply putting all these measures into a single subtitle is too complicated to read. Divide these measures into multiple subtitles. 2) Do not mix the R package information with the “measures” part. You can cite them in the “Data Analysis” part.

  2.5. Data Analysis

Line 262, page 6: DAGgity? Should the correct name be “DAGitty”?

Line 276, page 6: I believe you should use a period instead of a comma.

Line 258-288, page 6: 1) An intriguing figure, yet the connection between environmental noise and sleep habits is blurred. Using colors or dot lines might help. 2) What’s the difference between the mediating analysis and Directed Acyclic Graph analysis? If I understand correctly, is this a mediating model? If yes, why did you plot such a complicated figure?

3. Results

Line 304-314, page 7: 1) You have listed a ton of results in the texts, which one is the core point you want to present? 2) the results of Table 2 should be written in a separate paragraph. 3) I think the use of parentheses is incorrect. 4) Formatted fonts please.

Line 390-393, page 11: Hard to read and understand.

Line 395-409, page 11: 1) This paragraph looks like a “Discussion” to me. Reconsider the main outcome you want to show and revise this paragraph. The other paragraphs in the “Results” section contain the same problem, you may need to check them out. 2) Still, what’s the difference between the mediating analysis and Directed Acyclic Graph analysis?

Table 1, page 8: 1) Not an ideal layout to read, adjust the first column. 2) Unify all the digits please. 3) Where is the bottom line?

Table 2, page 9: 1) Not an ideal layout to read, adjust the legend row. 2) Overall mean exposure of Neighbourhood SES and Social class is confusing and misleading? What do you want to express using I to V? Shouldn’t I be the “highest” and V be the “lowest”?

Table 3, page 11: 1) Not an ideal layout to read, adjust the legend row. 2) Unified digits please.

Table 4, pages 11-12: 1) Not an ideal layout to read, adjust the legend row. 2) Formatted fonts please. 3) Unified digits please.

Figure 2, page 10: What’s the meaning of this figure?

Figure 3, page 10: Why the figure legend is separated? Did you check the formats before submitting?

4. Discussion

All of the paragraphs in this section are way too long. You may need to refine them.

I believe you have put a lot of effort into the revision process. However, the quality of this “manuscript” still has a giant gap to “publication” and revisions are still needed. This is an interesting subject and I hope you can make further revisions to meet the level for publication.

First, reading more papers from the TOP journals, to learn the formats, expressions, and of great importance—logic, might help a lot before revising. Second, most of the expressions contain redundancy and the writing logic is unclear. Logic plays a key role in scientific writing. Making your valuable data an intriguing story to tell and a coherent article for people to read, is of great importance. Read through all the texts and rephrase them. Third, the manuscript contains too many format errors and I really cannot point them out one by one, please adjust them. Last, finding a native English speaker to improve the writing can considerably improve the quality.

Thank you and my best,

Your reviewer

Author Response

REVIEWER 3

Please see the attachment where manuscript with suggested changes is attached.

Title: Environmental noise exposure and sleep habits among children in the INMA-Gipuzkoa cohort

Comment: major revision.

Detailed information:

Title: Writing an abbreviation in the title may be confusing.

Title has been changed and abbreviation has been removed from the title.

Abstract

  • The Abstract contains too much redundancy and you need to refine the main outcome of the study.

Rechecked and rewritten in text.

2) Line 21-23, page 1: Hard to read and understand.

Rephrased in text.

3) Line 30-32, page 1: Why did you write a Discussion-like sentence in the abstract? I suggest removing it.

Discussion-like sentence has been removed from the abstract.

4) Line 34, page 1: Writing an abbreviation in the title may be confusing.

Title has been changed for better understanding.

  1. Introduction

Line 49, page 2: I believe double quotations should be used instead of single quotations.

Corrected in text.

Line 54-112, pages 2-3: The logic of these paragraphs is confusing and vague. What do you really want to express? Which is the core scientific problem that needed to be addressed? I understand “Other factors” you wrote here is to coordinate with your GAP analysis yet this is not ideal. Writing a skeleton for your introduction before revising might help.

The idea of the paragraph is to understand all the factors (or as much as we can) explaining sleep in eleven-year old children and the interrelation between them. The answer to the question “How can environmental noise can affect sleep in children?” (the core scientific question that needed to be addressed) may involve other factors that impact  affecting environmental noise or sleep in children, as well as the interrelation between them. That is why a more complex theoretical framework is necessary and other factors affecting environmental noise or sleep in children should also be included in the question, as well as the relationships among them. That is why the presented DAG model, created based on the literature and revised by several experts in the field. We thought it was important to reflect this idea in the introduction. A introductory line to this paragraphs has been added to clear this idea.

Line 63-66, page 2: Are children participants to the same high noise level standards as adults?

As far as we know, no specific high noise level standards exists for children. According to EU definition high noise levels are noise levels above 55 dB Lden and 50 dB Lnight, with no specific regulation for vulnerable groups. However, as highlighted in the article, the number of studies regarding environmental noise health effects in children is small. Focussing in this topic, would allow to contribute information about standard values in this specific group

European Environment Agency. Environmental noise in Europe, 2020. EEA Report No 22/2019. 2020. Available from: https://op.europa.eu/publication/manifestation_identifier/PUB_THAL20003ENN

  1. Materials and Methods

2.1. Study population

Line 138-141, page 3: The inclusion criteria need to be clarified. Reading the “being older than 16 years” before the “the 11-year follow up” is dizzy.

Inclusion criteria was established for mothers at recruitment (during first trimester of pregnancy). As stated in the text “Since recruitment, data have been collected in several follow-up phases. In this study, we used data from the 11-year follow up”.  Thus, mothers were recruited during pregnancy following established criteria and their 11 year old children (as well as mothers themselves) are the participants in the 11-year follow up .

Line 141-142, page 3: Why did you choose the 11-year follow up? Any reason for it?

The main reason why 11-year old children follow-up was analysed is noise data availability at this age.  This study is part of INMA project (www.proyectoinma.org), which is a cohort study that consists on seven different study areas. The protocol and questionnaires for each follow-up data collection are defined among all the areas. Other environmental factors have been measured during different periods and environmental noise information was gathered in the 11-year-old follow-up.

Apart from the mentioned reason, this specific period (preadolescence) is considered especially important, due to the intense brain maturation (development of amygdala, hippocampus and prefrontal cortex) and cognitive development (Lupien et al., 2009; McEwen & Morrison, 2013; Simon et al., 2022; Stansfield & Clark, 2015). Taking into account the importance sleep plays during development, the study of the factors that could alter sleep in 11-year-old children is crucial. This could help the design of evidence-based preventive measures and health policies to protect against this potential problem.

Lupien, S.J., McEwen, B.S., Gunnar, M.R., Heim, C. Effects of stress throughout the lifespan on the brain, behaviour and cognition. Nat. Rev. Neurosci. 2009, 10, 434–445;  https://doi.org/10.1038/nrn2639

McEwen, B.S., Morrison, J.H.The Brain on Stress: Vulnerability and Plasticity of the Prefrontal Cortex over the Life Course. Neuron 2013, 79, 16–29. https://doi.org/10.1016/j.neuron.2013.06.028

Simon, K.R., Merz, E.C., He, X., Noble, K.G. Environmental noise, brain structure, and language development in children. Brain Lang2022, 229, 105112; https://doi.org/10.1016/J.BANDL.2022.105112

Stansfeld, S., Clark, C. Health Effects of Noise Exposure in Children. Curr. Environ. Heal. Reports 2015, 2, 171–178. https://doi.org/10.1007/s40572-015-0044-1

2.2. Environmental noise measurement

Line 160 and 162, page 4: What’s “façades”? Do you mean “facades”, the surface? If yes, what does this stand for?

Façade refers to the front of a building that faces on to a street or open space. As far as we know, both façade and facade are accepted. Therefore, noise measurements on the façade reflects the environmental noise immision levels in the outside of the building.

  2.3. Sleep measurement

Line 171-176, page 4: Is this a validated questionnaire? How do you ensure its validity and reliability?

The questionnaire used to asses sleep habits is an ad hoc questionnaire, shown in supplementary material (Appendix A1). The “Sleep Habits” comprises four questions, where bedtime, awakening  time and sleep habits are asked. Note that, as mentioned before, INMA project is a cohort study that consists on seven different study areas. The protocol and questionnaires for each follow-up data collection is defined among all the areas. This questionnaire was created and agreed among experts in the field from each study area in the INMA project and accepted by Ethics Committee from the hospital of each area.

  2.4. Other variables assessed

Line 193, page 4: Why the font style is different here? Ensure no format issues before submitting.

Format issues were checked carefully before submission. Mistake could derive from revision/edition process. However, we apologise for the errors and we will focus on this issue during later revision/resubmission to facilitate reviewer/editor work .

Line 189-248, pages 4-5: 1) Simply putting all these measures into a single subtitle is too complicated to read. Divide these measures into multiple subtitles. 2) Do not mix the R package information with the “measures” part. You can cite them in the “Data Analysis” part.

This section has been subdivided to make it easier to read. Regarding R package information, GGIR package was used to process data obtained from the actigraph. Therefore, it provides information about data creation or variable creation. That is why we believe this information is necessary when explaining the variable itself and fits in this section better than in the “Data Analysis” part.

  2.5. Data Analysis

Line 262, page 6: DAGgity? Should the correct name be “DAGitty”?

Corrected in text.

Line 276, page 6: I believe you should use a period instead of a comma.

Corrected in text.

Line 258-288, page 6: 1) An intriguing figure, yet the connection between environmental noise and sleep habits is blurred. Using colors or dot lines might help. 2) What’s the difference between the mediating analysis and Directed Acyclic Graph analysis? If I understand correctly, is this a mediating model? If yes, why did you plot such a complicated figure?

Image has been modified and added in text. While both mediation analysis and robust causal inference use DAGs, there are several differences between them. One main difference lies in the manner of estimating effects: whereas modern mediation analysis relies on counterfactuals, robust causal inference makes use of the principle of d-separation (to produce testable implications) and of adjustment criteria, such as the back-door criterion (to determine covariate adjustment sets for minimizing confounding bias). Our model, which has been scrutinized by means of testable implications, is not such an uncomplicated mediation model. Rather, it is a model composed of chains, forks, and colliders, which describes how quality of sleep occurs in nature: it is a model hypothesizing the data generation process. The complicated figure is needed because the process is complex.

Please see J. Pearl. 2009. Causality. Cambridge University Press 2 edn. DOI: 10.1017/CBO9780511803161.

  1. Results

Line 304-314, page 7: 1) You have listed a ton of results in the texts, which one is the core point you want to present? 2) the results of Table 2 should be written in a separate paragraph. 3) I think the use of parentheses is incorrect. 4) Formatted fonts please.

3.1 section was rechecked and rewritten: Information given in the Tables is not repeated in text, in order to present the most relevant results

Line 390-393, page 11: Hard to read and understand.

Rechecked and changed to have more precise and clear results.

Line 395-409, page 11: 1) This paragraph looks like a “Discussion” to me. Reconsider the main outcome you want to show and revise this paragraph. The other paragraphs in the “Results” section contain the same problem, you may need to check them out. 2) Still, what’s the difference between the mediating analysis and Directed Acyclic Graph analysis?

Section “3.4 The effect of environmental noise on sleep habits at 11 years of age” shows the results of the main objective of the study. In this paragraph, direct and total effects of environmental noise on sleep are shown, where no statistically significant effect was observed. Difference between the mediating analysis and DAG analysis is explained above.

Table 1, page 8: 1) Not an ideal layout to read, adjust the first column. 2) Unify all the digits please. 3) Where is the bottom line?

Table 1 format was edited and corrected.

Table 2, page 9: 1) Not an ideal layout to read, adjust the legend row. 2) Overall mean exposure of Neighbourhood SES and Social class is confusing and misleading? What do you want to express using I to V? Shouldn’t I be the “highest” and V be the “lowest”?

Legend row adjusted. Neighbourhood SES and Social class were classified in 5 groups, from highest to lowest. In the case of Neighbourhood SES, deprivation index was used: lower deprivation index, means higher socioeconomic status. Therefore, a lower number represents higher socioeconomic status. The variable Social Class was based in mothers´ occupation: a lower number represents a higher social class. Number I to V is just a way of ordering the different groups from higher to lower. This numbers has been changed to be more understandable.

Table 3, page 11: 1) Not an ideal layout to read, adjust the legend row. 2) Unified digits please.

Legend row adjusted and table format was changed to be more understandable.

Table 4, pages 11-12: 1) Not an ideal layout to read, adjust the legend row. 2) Formatted fonts please. 3) Unified digits please.

Legend row adjusted and table format was changed to be more understandable.

Figure 2, page 10: What’s the meaning of this figure?

Figure 2 was added in order to show noise exposition distribution. As environmental noise exposure description is one of the study objectives, we believe showing this distribution along with high environmental noise values is a visual way to be aware of the proportion of the sample exposed to noise exposition over recommended values.

Figure 3, page 10: Why the figure legend is separated? Did you check the formats before submitting?

 As explained before, format issues were checked carefully before submission and mistakes could derive from revision/edition process. However, we apologise for the errors and we will focus on this issue during later revision/resubmission to facilitate reviewer/editor work .

  1. Discussion

All of the paragraphs in this section are way too long. You may need to refine them .

We are aware of the length of the sentences and paragraphs but we believe, discussion is one of the strengths of the paper, and therefore, we believe all the information provided in the section is of vital importance.

I believe you have put a lot of effort into the revision process. However, the quality of this “manuscript” still has a giant gap to “publication” and revisions are still needed. This is an interesting subject and I hope you can make further revisions to meet the level for publication.

First, reading more papers from the TOP journals, to learn the formats, expressions, and of great importance—logic, might help a lot before revising. Second, most of the expressions contain redundancy and the writing logic is unclear. Logic plays a key role in scientific writing. Making your valuable data an intriguing story to tell and a coherent article for people to read, is of great importance. Read through all the texts and rephrase them. Third, the manuscript contains too many format errors and I really cannot point them out one by one, please adjust them. Last, finding a native English speaker to improve the writing can considerably improve the quality.

 We are grateful to your comments are article writing and format has been thoroughly rechecked after your recommendations. Revision of a English native speaker was performed after article submission, and second exhaustive revision has been before present resubmission. If needed, bills can be provided. Hope the article is more clear, accurate and intriguing after last changes.

Thank you and my best,

Your reviewer

Round 2

Reviewer 3 Report (New Reviewer)

ID: ijerph-2031333

Title: Environmental noise exposure and sleep habits among children in a cohort from northern Spain

I appreciate your efforts to improve the manuscript and to respond to the comments made in the first review process. Most problems were fixed yet there are still minor issues that need to be addressed.

Detailed information:

Title

Line 2, page 1: Why use the term “sleep habits”? I do not think it's the best term.

3. Results

Table 2, page 9: I do not know if the revised manuscript was checked but the legend row was still not aligned with the following rows.

Table 4, page 13: Formatted digits please. Use all numbers with 3-digit (e.g., -0.004 instead of - 0.0040) or any other forms you want to express. Just do not present numbers with 3-digit one place and 2-digit in another place.

Tables: The titles and notes of tables are mixed. Splitting them into titles and notes may be easier to read.

Besides the issues I mentioned above, no other further comments I have now.

Thank you and my best,

Your reviewer

Author Response

Please see the attachment, where latest version of the manuscript have been uploaded.

Title: Environmental noise exposure and sleep habits among children in a cohort from northern Spain

I appreciate your efforts to improve the manuscript and to respond to the comments made in the first review process. Most problems were fixed yet there are still minor issues that need to be addressed.

Detailed information:

Title

Line 2, page 1: Why use the term “sleep habits”? I do not think it's the best term.

The use of the term “sleep habits” was decided (instead of sleep quality) after a reviwer comment. The questionnaire used to asses sleep habits is shown in supplementary material (Appendix A1). Note that the questionnaire itself titles “Sleep Habits”. In this questionnaire, sleep habits in the past year are asked: usual bedtime and wake-up time, usual time to turn off the lights and usual time to fall asleep. We believe these questions refer to children´s sleep habits, as sleep habits involves behaviour pertaining to time to bed, time to rise, duration of night sleep, sleep latency and regularity of the mentioned ideas. Other studies also used the term “sleep habits” when referring to this constructs. Please see:

Arrona-Palacios, A.; Díaz-Morales, J.F.; Adan, A. Sleep Habits and Circadian Preferences in School-Aged Children Attending a Mexican Double-Shift School System. Sleep Med. 2021, 81, 116–119, doi:10.1016/j.sleep.2021.02.016.

Horiuchi, F.; Oka, Y.; Kawabe, K.; Ueno, S.I. Sleep Habits and Electronic Media Usage in Japanese Children: A Prospective Comparative Analysis of Preschoolers. Int. J. Environ. Res. Public Health 2020, 17, 1–12, doi:10.3390/ijerph17145189.

Horiuchi, F.; Kawabe, K.; Oka, Y.; Nakachi, K.; Hosokawa, R.; Ueno, S. Mental Health and Sleep Habits/Problems in Children Aged 3–4 Years: A Population Study. Biopsychosoc. Med. 2021, 15, 1–9, doi:10.1186/s13030-021-00213-2.

However, if other sleep terminology is thought to be more appropriate to the article publication, we will consider it. 

  1. Results

Table 2, page 9: I do not know if the revised manuscript was checked but the legend row was still not aligned with the following rows.

Legend row has been aligned.

Table 4, page 13: Formatted digits please. Use all numbers with 3-digit (e.g., -0.004 instead of - 0.0040) or any other forms you want to express. Just do not present numbers with 3-digit one place and 2-digit in another place.

Format errors have been fixed in Table 4.

Tables: The titles and notes of tables are mixed. Splitting them into titles and notes may be easier to read.

Titles and notes has been split to have easier read.

Besides the issues I mentioned above, no other further comments I have now.

Thank you and my best,

Your reviewer

This manuscript is a resubmission of an earlier submission. The following is a list of the peer review reports and author responses from that submission.

Round 1

Reviewer 1 Report

It is an important subject and a large sample size for a field experimental study. However, there are some necessary improvements that would increase quality of the article.

1) the introduction would benefit of also including the possibility of bidirectional associations between poor sleep and several outcomes, as for i.e cortisol and sleep 

2) please add in appendix or in the text the questions posed to measure sleep quality. Also consider the appropriateness of using the wording sleep quality if you only include sleep duration and sleep latency. That is not a common definition of sleep quality. I would say it that the use of sleep quality is confusing of not misleading the readers.  You also refer to ref 51 saying that this reference verifies that the use of actigraphy and questionnaires is deemed an efficient method for estimating sleep quality. Please revise and check for other references to motivate your use of method. Ref 51 only study a population with sleep disorders and furthermore only study one night (the first night with "the first night effect" is know to be affected in an experimental setting.

3) You will also find that actigraphy usually is found to overestimate sleep duration by 1-2 hours, and maybe also sleep efficiency. It could be wise to mention this in the discussion.

4) the level of hair cortisol can be confounded by physical exercise, hair dye, medication, maybe less relevant for this age group, but did you record and correct for these factors. I see that the level varied between 1,25 to 160 pg/mg, is this a normal variation for this age group?

5) I can not read out if there was a difference in exposure, or reported outcome among those wearing an actigraph or not?

6) how was early puberty dealt with?

7) please define what you dealt with different environmental noise sources, did you treat them as additive or multiple sources? I can not see any text on this

8. Figure 3 can not be read, the text is too small and slightly blurred. 

9) How did you correct for multi comparisons resulting in random significance as you included 165 tests? 

10) Table 4 and 5 are central in the result section, however table 4 seem to include an erroneous row break, or else I can not understand the single 0 on i.e 3rd row.  also you mention the trend, which trends do you refer to?

11) I am intrigued to know why you did not ask the 11 years old for their sleep report? 

Author Response

Response to Reviewer 1 Comments

It is an important subject and a large sample size for a field experimental study. However, there are some necessary improvements that would increase quality of the article.

Question 1: The introduction would benefit of also including the possibility of bidirectional associations between poor sleep and several outcomes, as for i.e cortisol and sleep 

Answer 1: Information has been included in text. We reconsidered the possibility of introducing the bidirectional associations in the introduction, as several presented variables are also affected by poor sleep quality. Nevertheless, bidirectional associations were not included in the created DAG model based on the purpose of the study. For example, effect of poor sleep on cortisol levels was not included because effect of physiological stress on sleep quality was prioritized.

Question 2:  Please add in appendix or in the text the questions posed to measure sleep quality. Also consider the appropriateness of using the wording sleep quality if you only include sleep duration and sleep latency. That is not a common definition of sleep quality. I would say it that the use of sleep quality is confusing of not misleading the readers.  You also refer to ref 51 saying that this reference verifies that the use of actigraphy and questionnaires is deemed an efficient method for estimating sleep quality. Please revise and check for other references to motivate your use of method. Ref 51 only study a population with sleep disorders and furthermore only study one night (the first night with "the first night effect" is know to be affected in an experimental setting.

Answer 2: Questions used to measure sleep quality have been added in appendix (Appendix A).

Regarding sleep quality definition, sleep quality does not only imply falling asleep and sleeping certain number of hours, but also implies other features, such as not awakening at nights, feeling rested, and self-satisfaction of sleep (Wang & Bíro, 2021). Therefore, we assume sleep quantity is also a feature of sleep quality and therefore sleep quality was used, as a general term. However, it is clarified in the methods section that sleep quality was estimated based on sleep quantity.  This idea was also included in the discussion.

Reference 51 was revised and changed with other reference in healthy children using actigraphy for 7 consecutive days and sleep questionnaires filled by parents.

Wang, F.; Bíró, É. Determinants of Sleep Quality in College Students: A Literature Review. Explore 2021, 17, 170–177, doi:10.1016/j.explore.2020.11.003.

Question 3:  You will also find that actigraphy usually is found to overestimate sleep duration by 1-2 hours, and maybe also sleep efficiency. It could be wise to mention this in the discussion.

Answer 3: As far as we know, literature states parents tend to overestimate sleep duration of their child; specifically they usually report earlier bedtimes and later wake-up times when comparing to actigraphic measures (Mazza et al., 2020; Werner et al., 2008). Parents’ overestimation of sleep duration varies in literature from 30 to 113 min per night, and this goes in line with our results. Discussion in the issue is now added in the text.

Mazza, S.; Bastuji, H.; Rey, A.E. Objective and Subjective Assessments of Sleep in Children: Comparison of Actigraphy, Sleep Diary Completed by Children and Parents’ Estimation. Front. Psychiatry 2020, 11, 1–11, doi:10.3389/fpsyt.2020.00495.

Werner, H.; Molinari, L.; Guyer, C.; Jenni, O.G. Agreement Rates between Actigraphy, Diary, and Questionnaire for Children’s Sleep Patterns. Arch. Pediatr. Adolesc. Med. 2008, 162, 350–358, doi:10.1001/archpedi.162.4.350.

Question 4:  The level of hair cortisol can be confounded by physical exercise, hair dye, medication, maybe less relevant for this age group, but did you record and correct for these factors. I see that the level varied between 1,25 to 160 pg/mg, is this a normal variation for this age group?

Answer 4: Some external factors might affect measurable levels of cortisol, including sun exposure, hair washing and hair bleaching products. Therefore, participants were asked not to wash their hair 48h prior hair collection and sample was not collected for those with died hair. Relation between physical activity and cortisol level was included in the DAG model (Figure 1).

Regarding cortisol level variations, studies about hair cortisol concentrations on children and adolescents are little and heterogeneous. Therefore, much is not known about reference intervals for hair cortisol in healthy children. However,  other studies with children and preadolescents also observed similar levels of hair cortisol concentration (Babarro et al., 2022; Gerber et al., 2017; Vanaelst et al., 2012) and wide ranges of cortisol levels (Anand et al., 2020; Hollenbach et al., 2019;). Note that cortisol concentration usually does not follow normal distribution, and logaritmic transformation is commonly ussed.

Anand, K., Rovnaghi, C. R., Rigdon, J., Qin, F., Tembulkar, S., Murphy, L. E., Barr, D. A., Gotlib, I. H., & Tylavsky, F. A. (2020). Demographic and psychosocial factors associated with hair cortisol concentrations in preschool children. Pediatric research87(6), 1119–1127. https://doi.org/10.1038/s41390-019-0691-2

Babarro, I., Ibarluzea, J., Theodorsson E., Fano E., Lebeña A., Guxens M., Sunyer J., & Andiarena A. (2022). Hair cortisol as a biomarker of chronic stress in preadolescents: influence of school context and bullying, Child Neuropsychology, DOI: 10.1080/09297049.2022.2115991

Gerber, M., Endes, K., Brand, S., Herrmann, C., Colledge, F., Donath, L., Faude, O., Pühse, U., Hanssen, H., & Zahner, L. (2017). In 6- to 8-year-old children, hair cortisol is associated with body mass index and somatic complaints, but not with stress, health-related quality of life, blood pressure, retinal vessel diameters, and cardiorespiratory fitness. Psychoneuroendocrinology, 76, 1–10. https://doi.org/10.1016/j.psyneuen.2016.11.008

Hollenbach, J. P., Kuo, C. L., Mu, J., Gerrard, M., Gherlone, N., Sylvester, F., Ojukwu, M., & Cloutier, M. M. (2019). Hair cortisol, perceived stress, and social support in mother-child dyads living in an urban neighborhood. Stress (Amsterdam, Netherlands)22(6), 632–639. https://doi.org/10.1080/10253890.2019.1604667

Vanaelst, B., De Vriendt, T., Huybrechts, I., Rinaldi, S., & De Henauw, S. (2012). Epidemiological approaches to measure childhood stress. Paediatric and Perinatal Epidemiology, 26(3), 280–297. https://doi.org/10.1111/j.1365-3016.2012.01258.x

Question 5: I can not read out if there was a difference in exposure, or reported outcome among those wearing an actigraph or not?

Answer 5: Comparisons between two groups (group wearing an actigraph and group not wearing it) were performed. No differences were observed between two groups in environmental noise exposure variables or reported sleep variables. This information is added in the text.

Question 6:  How was early puberty dealt with?

Answer 6:  Early puberty is defined as the apparition of puberty signs before 8 years of age for girls and 9 years of age for boys (Res, 2009). Early puberty and puberty stage  has been associated with sleep quality, usually sleep duration being smaller and bed timing later among those with advanced puberty (Diao et al., 2020; Luciet et al., 2021).

Tanner scale was filled by parents in the 11 year follow-up. In our sample, 93,6% of the boys remain in the T1 and T2 Tanner stages, and 90,7% of the girls below T3 stages (among them %70,6 remain in T1 or T2 stages). Studies show that at 11 years, most of the children stay in T1 and T2 stages (Konforte et al., 2013; Miller et al., 2020). Therefore, considering the age mean of our sample was around 11 years old and that each Tanner stage lasts around 12-15 months in normal white European puberty, we assume that the proportion of early puberty in our sample was small.

Even though, we compared if there is any significant difference in any sleep outcome considered between prepuberal (those in T3, T4 or T5) and non-prepuberal (those remaining in T1 or T2 stages) subjects. No significant differences were observed among two groups in any sleep outcome. Therefore, early puberty was not considered in the present study.

Diao, H.; Wang, H.; Yang, L.; Li, T. The Association between Sleep Duration, Bedtimes, and Early Pubertal Timing among Chinese Adolescents: A Cross-Sectional Study. Environ. Health Prev. Med. 2020, 25, 1–8. https://doi.org/10.1373/10.1186/s12199-020-00861-w

Konforte, D., Shea, J.L., Kyriakopoulou, L., Colantonio, D., Cohen, A.H., Shaw, J., Bailey, D., Chan, M.K., Armbruster, D., Adeli, K., 2013. Complex biological pattern of fertility hormones in children and adolescents: A study of healthy children from the CALIPER cohort and establishment of pediatric reference intervals. Clin. Chem. 59, 1215–1227. https://doi.org/10.1373/clinchem.2013.204123

Lucien, J.N., Ortega, M.T., Shaw, N.D., 2021. Sleep and puberty. Curr. Opin. Endocr. Metab. Res. 17, 1–7. https://doi.org/10.1016/j.coemr.2020.09.009

Miller, B.S., Sarafoglou, K., Yaw Addo, O., 2020. Development of tanner stage-age adjusted CDC height curves for research and clinical applications. J. Endocr. Soc. 4, 1–13. https://doi.org/10.1210/jendso/bvaa098

Res, J.C.; Endo, P. Precocious Puberty and Normal Variant Puberty : Definition , Etiology , Diagnosis And. 2009, 1, 164–174, https://doi.org/10.4274/jcrpe.v1i4.3

Question 7:  Please define what you dealt with different environmental noise sources, did you treat them as additive or multiple sources? I can not see any text on this

Answer 7: Different sources of environmental noise were added. This information is now available in text.

Question 8:  Figure 3 can not be read, the text is too small and slightly blurred.

Answer 8: Edited in text.

Question 9:  How did you correct for multi comparisons resulting in random significance as you included 165 tests? 

Answer 9: Under the present statistical approach, it is not recommended to correct for multiple comparisons. Thus, we did not apply any of these procedures (such as the Bonferroni correction).

We do understand why the reviewer asks this question. Nonetheless, the reason why we did not apply any of these procedures is that the classic issue of multiple comparisons is not, in fact, present in our statistical problem/approach (testing the validity of a given DAG via the so-called testable implications). This is so because the typical issue of multiple comparisons arises when we try to simultaneously assess/test the association of a certain single exposure (or randomized treatment) with many hypothesized responses. Testing the validity of a DAG via testable implications is not the same issue as the problem of multiple comparisons, even if many tests of hypothesis occur simultaneously in both cases. 

Question 10:  Table 4 and 5 are central in the result section, however table 4 seem to include an erroneous row break, or else I can not understand the single 0 on i.e 3rd row.  also you mention the trend, which trends do you refer to?

Answer 10: Regarding the table, there was a format error, which has been corrected in text.

Trend is referred to the effect (direct and total) of environmental noise (both evening and nocturnal noise) on subjective sleep quality. Although non significant, in all cases a higher environmental noise has an positive effect on reported nocturnal sleep and a negative effect on reported sleep latency. This is the reason we refer a tendency.

Question 11:  I am intrigued to know why you did not ask the 11 years old for their sleep report? 

Answer 11: INMA project is a cohort study that consists on seven different study areas. The protocol for each follow up data collection is defined among all the areas in order to obtain an agreed protocol. In this case, the main reason why 11 years old were not asked about their sleep report was not to overload them; it was decided that neuropsychological tests and questionnaires about general well-being and relationship with partners would be completed by children. Parents completed another questionnaire about their children. Note that adolescents themselves completed the sleep report in the 14 year follow-up.

Please see the attachment: Article was attached with the suggested changes.

Reviewer 2 Report

     I question the definition of noise - used to be described as unwanted and intrusive sounds but now is generally described as harmful sounds.  Cannot understand why noise is defined as coming from "human activity." While humans may have control over construction noise and road traffic noise, I would not generally call these sounds "human activity" sounds.  Do rethink definition.

    I assume your measurements are dBA - should not simply state dB.  Do recheck this.

     I would have a section entitled Subjects before Methods are discussed,  You describe subjects in Results section.  Believe this should be introduced earlier.  Also you mention questionnaires but do not show them to your readers.  Believe questionnaires should be shown to readers.

     You did not discuss impacts of aircraft noise on children's sleep.  Would suggest you look at 2021 study by Lee, et al.   With so much discussion about impacts on children in European countries, one might wonder whether results of this study could be generalized and useful to researchers in Canada, Asia, United States, etc.

     I appreciate that the authors recognized and discussed the limitations of their results, especially with regard to groups of different social status, and suggested that more research in this area is called for.  Yes, larger urban areas and areas impacted by aircraft noise should be studied,  However, this study is deserving of publication as we move forward to a better understanding of the impacts of noise on children's sleep and overall physical and mental health.

Author Response

Response to Reviewer 2 Comments

Question 1:   I question the definition of noise - used to be described as unwanted and intrusive sounds but now is generally described as harmful sounds.  Cannot understand why noise is defined as coming from "human activity." While humans may have control over construction noise and road traffic noise, I would not generally call these sounds "human activity" sounds.  Do rethink definition.

Answer 1: Definition was based on the one proposed in the The Environmental Noise Directive (Directive 2002/49/EC), where environmental noise is defined as: unwanted or harmful outdoor sound created by human activities, including noise emitted by means of transport, road traffic, rail traffic, air traffic, and from sites of industrial activity. The World Health Organization (WHO) defines environmental noise as noise from all sources with the exception of workplace noise being noise all unwanted sound or set of sounds that causes annoyance or can have a health impact. After reconsidering definition, “any sound derived from human activity that is unpleasant, unwanted or harmful” may fit better. Edited in text.

  Question 2:    I assume your measurements are dBA - should not simply state dB.  Do recheck this.

Answer 2: Definition of day-evening-night level (Lden) proposed in the  Directive 2002/49/EC of the European Parliament and of the Council of 25 June 2002 relating to the assessment and management of environmental noise states that Lden unit is dB. However, Lday, Levening and Lnight present in the formula are A-weighted long-term average sound level and therefore should be stated in dBA. Rechecked in the text.

    Question 3:  I would have a section entitled Subjects before Methods are discussed,  You describe subjects in Results section.  Believe this should be introduced earlier.  Also you mention questionnaires but do not show them to your readers.  Believe questionnaires should be shown to readers.

Answer 3:  Description of the sample have been moved into methods, where participants´characteristics are described. Questions posed to measure sleep quality have been added in appendix (Appendix A).

    Question 4:   You did not discuss impacts of aircraft noise on children's sleep.  Would suggest you look at 2021 study by Lee, et al.   With so much discussion about impacts on children in European countries, one might wonder whether results of this study could be generalized and useful to researchers in Canada, Asia, United States, etc.

Answer 4:  The discussion about impacts of aircraft noise on children's sleep was added in the text. Regarding the question if results of the study could be generalizable and useful to non-European researchers, the amount of studies about environmental noise effects on children´s sleep is little, with heterogeneous methodology and most of them European. Therefore, we do not have enough support to generalization of the results to non-European countries. However, we believe the need of further research in the issue, with standardized methodologies in order to create generalizable conclusions.

    I appreciate that the authors recognized and discussed the limitations of their results, especially with regard to groups of different social status, and suggested that more research in this area is called for.  Yes, larger urban areas and areas impacted by aircraft noise should be studied,  However, this study is deserving of publication as we move forward to a better understanding of the impacts of noise on children's sleep and overall physical and mental health.

Please see the attachment: Article was attached with the suggested changes
